# Umbilical-Cord-Derived Mesenchymal Stromal Cells Modulate 26 Out of 41 T Cell Subsets from Systemic Sclerosis Patients

**DOI:** 10.3390/biomedicines11051329

**Published:** 2023-04-30

**Authors:** Paula Laranjeira, Francisco dos Santos, Maria João Salvador, Irina N. Simões, Carla M. P. Cardoso, Bárbara M. Silva, Helena Henriques-Antunes, Luísa Corte-Real, Sofia Couceiro, Filipa Monteiro, Carolina Santos, Tânia Santiago, José A. P. da Silva, Artur Paiva

**Affiliations:** 1Flow Cytometry Unit, Department of Clinical Pathology, Centro Hospitalar e Universitário de Coimbra, 3000-075 Coimbra, Portugal; 1979paula@gmail.com; 2Coimbra Institute for Clinical and Biomedical Research (iCBR), Faculty of Medicine, University of Coimbra, 3000-548 Coimbra, Portugal; jdasilva@ci.uc.pt; 3Center for Innovative Biomedicine and Biotechnology (CIBB), University of Coimbra, 3000-548 Coimbra, Portugal; 4Center for Neuroscience and Cell Biology (CNC), University of Coimbra, 3004-504 Coimbra, Portugal; 5Stemlab S.A., Famicord Group, 3060-197 Cantanhede, Portugal; francisco.santos@crioestaminal.pt (F.d.S.); irina.simoes@crioestaminal.pt (I.N.S.); carla.cardoso@crioestaminal.pt (C.M.P.C.); helena.antunes@crioestaminal.pt (H.H.-A.); luisagacr@gmail.com (L.C.-R.); sofia.couceiro@crioestaminal.pt (S.C.); filipa.monteiro@crioestaminal.pt (F.M.); carolina.santos@crioestaminal.pt (C.S.); 6Rheumatology Department, Hospitais da Universidade de Coimbra, Centro Hospitalar e Universitário de Coimbra, 3000-075 Coimbra, Portugal; mjsalvadorhenriques@gmail.com (M.J.S.); tlousasantiago@hotmail.com (T.S.); 7Algarve Biomedical Center (ABC), Universidade do Algarve, 8005-139 Faro, Portugal; barbarasmsilva@gmail.com; 8Algarve Biomedical Center Research Institute (ABC-RI), Universidade do Algarve, 8005-139 Faro, Portugal; 9Doctoral Program in Biomedical Sciences, Faculty of Medicine and Biomedical Sciences, Universidade do Algarve, 8005-139 Faro, Portugal; 10Instituto Politécnico de Coimbra, ESTESC-Coimbra Health School, Ciências Biomédicas Laboratoriais, 3046-854 Coimbra, Portugal

**Keywords:** mesenchymal stromal cells (MSCs), mesenchymal stem cells, immunomodulatory potential, systemic sclerosis, T cells, T cell polarization, T cell activation, Treg, Th17, cellular therapy

## Abstract

Systemic sclerosis (SSc) is an immune-mediated disease wherein T cells are particularly implicated, presenting a poor prognosis and limited therapeutic options. Thus, mesenchymal-stem/stromal-cell (MSC)-based therapies can be of great benefit to SSc patients given their immunomodulatory, anti-fibrotic, and pro-angiogenic potential, which is associated with low toxicity. In this study, peripheral blood mononuclear cells from healthy individuals (HC, *n* = 6) and SSc patients (*n* = 9) were co-cultured with MSCs in order to assess how MSCs affected the activation and polarization of 58 different T cell subsets, including Th1, Th17, and Treg. It was found that MSCs downregulated the activation of 26 out of the 41 T cell subsets identified within CD4^+^, CD8^+^, CD4^+^CD8^+^, CD4^−^CD8^−^, and γδ T cells in SSc patients (HC: 29/42) and affected the polarization of 13 out of 58 T cell subsets in SSc patients (HC: 22/64). Interestingly, SSc patients displayed some T cell subsets with an increased activation status and MSCs were able to downregulate all of them. This study provides a wide-ranging perspective of how MSCs affect T cells, including minor subsets. The ability to inhibit the activation and modulate the polarization of several T cell subsets, including those implicated in SSc’s pathogenesis, further supports the potential of MSC-based therapies to regulate T cells in a disease whose onset/development may be due to immune system’s malfunction.

## 1. Introduction

Systemic sclerosis (SSc), or scleroderma, is an immune-mediated disease characterized by vasculopathy and fibrosis of the skin and internal organs, which affects the lungs, heart, and gastrointestinal tract. The clinical course of this rheumatic disease is associated with a poor quality of life and a high mortality rate [1,2,3,4].

Deep alterations are found in the innate and acquired immune system in SSc patients, including the presence of autoantibodies, which are primarily directed against nuclear and nucleolar antigens and detected in the majority of patients. Indeed, an autoimmune reaction to endothelial cells is believed to be caused by the initiation of SSc, which leads to endothelial cell damage and subsequent vasculopathy and tissue fibrosis [1,5,6,7,8,9]. The apoptosis of endothelial cells can precipitate the defective vasomotor regulation (which controls blood vessel constriction and dilatation) verified in cases of SSc, leading to inefficient blood supply to the tissues and further promoting endothelial cell death [6]. Of note, the compensatory mechanisms that promote angiogenesis in hypoxic conditions also seem to fail in SSc patients, which is possibly due to defects in endothelial progenitor cells [6]. Besides apoptosis, other pathological mechanisms affecting endothelial cells have been described in cases of SSc, including endothelial-to-mesenchymal transdifferentiation and endothelial cell activation [7,8]. Endothelial-to-mesenchymal transdifferentiation can be induced by inflammatory mediators, autoantibodies directed against endothelial cells, or hypoxia; in these conditions, SSc endothelial cells acquire a myofibroblast phenotype and the ability to produce collagen [8]. Likewise, endothelial cell activation results in the upregulation of chemotactic factors, the vasoconstriction of blood vessels, and subendothelial fibrosis, leading to the proliferation of blood vessels’ muscular layers and intraluminal thrombosis, with consequent tissue hypoxia. Finally, endothelial cell apoptosis leads to the destruction of the blood vessels, also resulting in tissue hypoxia [8]. As previously referred, an inappropriate immune response against endothelial cells can be precipitated by the onset of SSc. In turn, apoptotic endothelial cells recruit and activate immune cells, establishing a positive feedback loop.

In SSc, skin fibrotic lesions are preceded by the establishment of a rich T cell inflammatory infiltrate, highlighting the role of T cells in the disease’s development. Both circulating and fibrotic tissue-infiltrating SSc immune cells display a type I interferon (IFN) signature and a bias towards Th2 polarization, resulting in high levels of interleukin (IL)-4, IL-13, and IL-31 which, together with IL-1β, IL-6, and transforming growth factor (TGF)-β, induce fibroblasts to proliferate and synthesize an excessive amount of extracellular matrix (ECM), further contributing to tissue fibrosis [1,3,4,5,7,8,10]. In fact, fibroblasts from SSc patients present an activated phenotype, which is associated with an enhanced production of ECM components and the expression of α-smooth muscle actin (α-SMA, a myofibroblast marker). As all these features are retained ex vivo, they are likely the result of genetic or epigenetic factors stemming from fibroblasts [4,6,7]. Myofibroblasts are considered a transient cell population that can arise from different cell types (such as bone marrow stem cells, monocytes/fibrocytes, pre-adipocytes and adipocytes, fibroblasts, pericytes endothelial cells, and epithelial cells) with important physiological functions, such as tissue repair [3,4,6,8]. In SSc, myofibroblasts are expanded, hyper-functional, and the main producers of ECM, including type I and type III collagen and fibronectin [3,4,8]. TGF-β assumes critical importance in the initiation and maintenance of the fibrogenic response induced by SSc fibroblasts and myofibroblasts. This cytokine upregulates the expression of different types of collagen and other ECM proteins, induces the transdifferentiation of fibroblasts into myofibroblasts, and downregulates matrix metalloproteinase (MMP) expression [4,6,7,11]. In addition to the higher levels of TGF-β observed in SSc, SSc fibroblasts present an increased expression of type I and type II TGF-β receptors along with alterations in TGF-β pathways, which render them hyperresponsive to TGF-β, further enhancing their pro-fibrotic response [6,7]. Other factors that control ECM deposition are also being investigated, such as mutations in the ECM glycoprotein fibrillin-1 (driving alterations in the organization of the ECM and the increased secretion of other ECM components) [6] and the role of MMPs and tissue inhibitors of metalloproteinases (TIMPs) [11]. MMPs are proteases with the ability to degrade ECM components and regulate tissue fibrosis and remodeling. TIMPs are inhibitors of MMP activity. MMP7 was found to be present at increased concentrations in serum extracted from SSc patients, especially those with pulmonary involvement, and was positively correlated with lung dysfunction [12]. In turn, lower levels of MMP-1 mRNA were found in both affected and unaffected skin samples from SSc patients, accompanied by an increased number of TIMP-1 transcripts in the skin [11] and higher TIMP-1 protein levels in the serum [13]. Importantly, MMP-1 and MMP-3 polymorphisms are associated with interstitial lung disease and reduced diffusing capacity for carbon monoxide, respectively [14]. All these factors that contribute to the exacerbation of the fibrogenic response in SSc ultimately result in an unrestrained deposition of ECM and tissue remodeling, culminating in the loss of function of the affected organs.

Besides Th2, several other T cell subsets are implicated in the pathogenesis of SSc, such as CD4^+^CD8^+^ T cells, CD8^+^ T cells, cytotoxic CD4^+^ T cells, follicular helper T (Tfh) cells, Treg, Th17, Th9, Th22, γδ T cells, and angiogenic T cells (Tang) cells, which are identified as CD3^+^CD31^+^CXCR4^+^. These T cell subsets can induce endothelial cell apoptosis, release pro-inflammatory and pro-fibrotic cytokines (IL-1β, IL-4, IL-5, IL-13, IL-17, IL-21, IL-31 TGF-β), or induce fibroblasts activation and transdifferentiation into myofibroblasts [3,5,10,15,16,17,18,19,20,21,22,23]. In turn, SSc fibroblasts and myofibroblasts produce immune mediators that shape the immune response, as in the case of IL-33, for instance, a pro-fibrotic cytokine with the ability to induce Treg differentiation into Th2-like cells [5,17,20].

Conventional treatments involve immunotherapy, which shows limited benefit for SSc patients. Thus, despite the associated risks, autologous hematopoietic stem cell transplantation is still the most effective treatment for SSc [1,20], demonstrating the importance of the adaptive immune system in this disease and fostering research into new therapeutic agents targeting immune cells. In this context, mesenchymal stromal cells (MSCs, which comprise mesenchymal stem cells) can be a privileged candidate for treating SSc. The diversity of T cell subsets involved in SSc pathogenesis is considerable and, according to our previous experience, MSCs’ immunomodulatory activity over T cell function is transversal with respect to several T cell subsets, both in health and immune-mediated diseases [24,25,26,27]. The simultaneous immunomodulatory, anti-fibrotic, and pro-angiogenic action of MSCs, which is associated with low toxicity [28,29], renders MSC-based therapy a privileged approach for SSc treatment. Nevertheless, studies focusing on the immunomodulatory effect of MSCs on T cells from SSc patients are scarce. MSC-based therapy can be a game-changer, especially for SSc with an aggressive course or patients refractory to conventional immunomodulatory therapies.

In SSc animal models, MSC infusion reduced pulmonary fibrosis and inflammation and improved skin fibrosis and ulcers [29,30]. These pre-clinical studies are uncovering the molecular mechanisms MSCs may use to regulate the immune systems (hepatocyte growth factor (HGF) and IL-1-RA), fibrosis (HGF, miR-29a-3p, and miR-151-5p, among which the latter downregulates the IL-4R pathway), and vasculopathy (vascular endothelial growth factor (VEGF)) of SSc patients [29,31]. Accordingly, the pro-angiogenic effect of MSCs was also demonstrated after focal cerebral ischemia in mice, wherein MSC-derived small extracellular vesicles fostered endothelial migration, proliferation, and tube formation in vitro and promoted angiogenesis in vivo [32].

A search of ClinicalTrials.gov (https://clinicaltrials.gov), accessed on 6 February 2023, led to the retrieval results of eight phase I/II clinical trials, among which six clinical trials are using allogeneic mesenchymal stem cells that have been isolated from umbilical cords or bone marrow and two clinical trials are using autologous adipose-tissue-derived stromal vascular fraction (AD-SVF), which is rich in mesenchymal stem cells. Of note, no severe adverse events were registered in the clinical trials over a 6- or 12-month follow-up period [33,34,35,36,37].

In small cohorts of SSc patients (*n* < 20), the local infusion of autologous AD-SVF to treat hand disability did not yield results that were statistically different from a placebo, but it improved the healing of digital ulcers and prevented the development of new ulcers [35,38]. Noteworthy, several studies have indicated important alterations in the MSCs from SSc patients, which may explain the unsatisfactory results obtained with autologous AD-SVF. SSc MSCs present a deficient capacity to differentiate into endothelial cells, early senescence, reduced proliferation rates, decreased migratory ability, and reduced immunosuppressive potential [29,39,40]. Furthermore, SSc MSCs display a pro-fibrotic propensity [29,41,42], and the exposure of dermal MSCs from healthy individuals to an SSc dermal environment induces the expression of myofibroblast-commitment genes while reducing the expression of vascular repair genes [43].

Notwithstanding, the few published data from case reports and clinical trials carried out with allogeneic MSCs show promising results, namely, a significant clinical improvement in digital ulceration, vasculopathy, and skin fibrosis, after bone marrow-derived MSC administration [29,36,44,45,46]. A combined therapy consisting of plasmapheresis, cyclophosphamide, and allogeneic umbilical-cord-derived MSCs applied to a cohort of 14 SSc patients resulted in decreased skin fibrosis, the size and pain of digital ulcers, and improved lung function, which was accompanied by a decrease in the serum levels of TGF-β and anti-Scl70 autoantibodies in a 12-month follow-up [34,37].

Other clinical trials directly targeting immune cells implicated in SSc pathophysiology are in progress. For instance, clinical trials targeting B cells with Rituximab have indicated its safety and efficacy with respect to improving skin and pulmonary function [47,48].

In order to better understand the effect of MSCs, particularly MSCs derived from umbilical cord tissue, on T cells from SSc patients, we identified T cell subsets with great detail and analyzed umbilical-cord-derived MSCs’ effects on each one of the 57 T cell subsets identified. Accordingly, peripheral blood mononuclear cells (PBMCs) from healthy individuals and SSc patients were cultured in the absence/presence of umbilical-cord-derived MSCs and in the absence/presence of T-cell-stimulating agents. We focused our analysis on how the presence of MSCs affected T cell activation and polarization. For this purpose, SLCTmsc02, an investigational Advanced Therapy Medicinal Product (iATMP) based on umbilical-cord-derived MSCs that was developed and manufactured by Stemlab S.A., was used.

## 2. Materials and Methods

### 2.1. Study Population

This study enrolled 9 unselected SSc patients (9 women and 0 men, ranging from 33 to 82 years with a mean age: 57 ± 17 years) followed at the Rheumatology Department, Centro Hospitalar e Universitário de Coimbra, Portugal. All patients fulfilled the 2013 American College of Rheumatology (ACR)/European League against Rheumatism (EULAR) classification criteria for SSc. Demographic and clinical data of the patients are described in Table 1. Patients below 18 years of age or who were pregnant or afflicted with an active infectious disease, cancer, or an associated auto-immune disease were excluded from the study. A control group of 6 healthy individuals (HC), comprising 5 women and 1 man with ages ranging from 35 to 58 years (mean age: 46 ± 8.0 years), was also included in the study. All participants were enrolled in the study from June 2021 to March 2022. All subjects gave their written informed consent for inclusion before they participated in the study. The study was conducted in accordance with the Declaration of Helsinki, and the protocol was approved by the Ethics Committee of Centro Hospitalar e Universitário de Coimbra (Ref.: CHUC-202-20).

### 2.2. GMP Manufacturing of SLCTmsc02–Umbilical Cord Tissue MSCs

The starting material for SLCTmsc02 is an umbilical cord tissue cell suspension cryopreserved at Stemlab S.A. cell bank (Cantanhede, Portugal). The Cryopreservation Laboratory of Stemlab S.A. (Cantanhede, Portugal) is a facility that has been authorized by the national competent authority Portugal’s Directorate-General of Health (Direção-Geral da Saúde, DGS) to perform analysis, processing, storage, and distribution activities related to umbilical cord tissue. Additionally, Stemlab S.A. has been accredited by the Association for the Advancement of Blood & Biotherapies (AABB), according to the Standards for Cellular Therapy Product Services, for the processing, storage, and distribution of somatic cells (MSCs from umbilical cord tissue). Upon the reception of the starting material, the Head of Quality Control verifies its conformance according to the established specifications and releases it to produce the SLCTmsc02 under GMP conditions.

The cell suspension was thawed, and cells were cultured for two passages to allow the enrichment of the MSC population. Upon reaching a confluence level of approximately 80%, cells were washed (DPBS, Gibco, Pasley, UK), detached (TrypLE, Gibco, Pasley, UK), and resuspended in complete cell culture medium for further expansion. Then, cells were detached and resuspended in 5% human serum albumin (Albunorm 5%, Octapharma, Lachen switzerland). After analytical characterization, a cryopreserving solution was added (CryoSure-DEX40, WAK-Chemie Medical GmbH, Steinbach, Germany), and the cells were cryopreserved as an Intermediate Product (IP) until request for production of final product units.

When required, IP units were thawed and cells were cultured for one more passage. Upon reaching a confluence level of approximately 80%, cells were detached and SLCTmsc02 units were formulated and cryopreserved.

SLCTmsc02 units were subjected to a complete quality control assessment for identity and purity (cell number, viability, and immunophenotype), safety (sterility, endotoxins, mycoplasma, and karyotype), and potency (IDO production).

### 2.3. Peripheral Blood Mononuclear Cell Isolation

Lymphoprep (Stemcell Technologies, Vancouver, BC, Canada) gradient density centrifugation was used for the isolation of peripheral blood mononuclear cells (PBMCs) from heparin-collected peripheral venous blood; this procedure was performed according to the manufacturer’s instructions and as previously described by our group [50]. PBMCs were washed twice in Hank’s Balanced Salt Solution (HBSS, Gibco, Life Technologies, Paisley, UK) and resuspended in RPMI 1640 supplemented with GlutaMax medium (Invitrogen, Life Technologies, Waltham, MA, USA), 10% fetal bovine serum (FBS), and antibiotic-antimycotic (Gibco, 100 units/mL of penicillin, 100 µg/mL of streptomycin, and 0.25 µg/mL of amphotericin B).

### 2.4. Cell Culture

PBMCs from SSc patients and healthy individuals (HC) were cultured in the presence/absence of MSCs and presence/absence of the T cell mitogen phytohemagglutinin (PHA, Fujifilm Irvine Scientific, Inc., Santa Ana, CA, USA). To complete this procedure, in cell culture plates (Falcon, Becton Dickinson Biosciences (BD), San Jose, CA, USA) and always in a final volume of 1 mL of RPMI 1640 medium with GlutaMax and 10% FBS, 1 × 10^6^ PBMCs were plated in the presence/absence of 0.1 × 10^6^ MSCs (establishing a PBMC:MSC ratio of 10:1) and in the presence/absence of PHA (1 μg/mL). Cells were cultured for 96 h at 37 °C in a sterile and humidified atmosphere containing 5% CO_2_. Whenever appropriated, PHA was added 72 h after the beginning of the cell culture, thereby stimulating T cells for a 24 h period. Our experimental procedure is schematized in Figure 1.

### 2.5. Flow Cytometry Analyses

#### 2.5.1. Staining Protocol and Sample Acquisition

The contents of each well of the cell culture plate were transferred to a 5 mL polystyrene tube (12 × 75 mm) and centrifuged for 5 min (540× *g*). The supernatant was discarded, and the cell pellet was stained with the monoclonal antibodies (mAb) described in detail in Table 2. After a 10 min incubation period in the dark and at room temperature, cells were fixed via 10 min incubation with 1 mL of FACSLysing Solution (BD). Cells were centrifuged (540× *g*, 5 min) to discard the FACSLysing Solution and washed with 1 mL of Dulbecco’s phosphate-buffered saline (PBS, Corning, Manassa, VA, USA). Finally, the cell pellet was resuspended in 500 µL of PBS and immediately acquired in a FACSLyric flow cytometer (BD) using FACSuite acquisition software (v1.5.0.925, BD, San Jose, CA, USA).

#### 2.5.2. Gating Strategy to Identify T Cell Subsets

For this study, we have specifically designed an mAb panel (Table 2) that enables the identification of 80 different T cell subsets. However, given the reduced number of some T cell subsets, we were only able to accurately analyze 64 T cell subsets. Accordingly, the following gating strategy was used: after the exclusion of doublets and cell debris (based on FSC-A and FSC-H properties), T cells were identified as CD3^+^ events, with characteristic FSC-A and SSC-A properties. T cells were divided into TCRαβ T cells and TCRγδ T cells based on TCRγδ expression. According to their CD4 and CD8 expression, CD4^+^CD8^−^, CD4^−^CD8^+^, CD4^+^CD8^+^, and CD4^−^CD8^−^ TCRαβ T cells were defined. Within each one of these five major T cell subpopulations (four TCRαβ T cell subsets and γδ T cells), we proceed to identify CD25^++^CD127^low/−^ regulatory T cells (Treg), CXCR5^+^ follicular T cells (Tf), and the remaining (non-Treg CXCR5^−^) T cells. Treg cells were further subdivided into follicular regulatory cells (Tfr, corresponding to CXCR5^+^ Treg cells) and CXCR5-negative Treg. Four other subpopulations were defined within each one of these twenty T cell subsets according to the expression pattern of CXCR3 and CCR6: CXCR3^+^CCR6^−^ (T1), CXCR3^−^CCR6^+^ (T17), CXCR3^+^CCR6^+^ (T1/17), and CXCR3^−^CCR6^−^. This latter T cell subset corresponds to T cells that are neither T1- nor T17-polarized; thus, the vast majority of these CXCR3^−^CCR6^−^ T cells were T0 and T2-polarized T cells. Finally, in each one of the eighty T cell subsets identified up to this point, the percentage of early-activated (CD25^+^) or late-activated (HLA-DR^+^) T cells was evaluated. Overall activation was also assessed in the major subpopulations. A scheme displaying all the T cell subsets studied can be found in Figure 2. The flow cytometry dot plot histograms exemplifying the gating strategy used are depicted in Figure 3. The aim of this study was to evaluate the effect of MSCs on T cell polarization and activation, in each one of the eighty subsets, with respect to both HC and SSc. Due to the low levels of representation of some T cell subsets, we were able to study T cell polarization in 64 subsets and T cell activation in 42 subsets.

### 2.6. Statistical Analysis

Mean and standard deviation were calculated for all variables under study. Differences between HC and SSc were evaluated using the Mann–Whitney test. To compare different culture conditions, with respect to either HC or SSc, Friedman test and Wilcoxon test were applied whenever appropriate. Statistical analysis was performed with Statistical Package for Social Sciences (IBM SPSS, version 27, Armonk, NY, USA) software. Differences were considered statistically significant whenever *p* < 0.05.

## 3. Results

Aiming to uncover the larger picture with respect to how MSCs influence different T cell subsets, we carried out an experimental design that enabled the identification of 64 different T cell subsets and the simultaneous evaluation of the effect of MSCs on each individual subset. As summarized in Table 3, MSCs’ ability to inhibit T cell activation was transversal with respect to several T cell subsets and more marked in the later stages of T cell activation, as assessed by the percentage of T cells expressing the late-activation marker HLA-DR. MSCs inhibited T cell activation for both HC and SSc. In the same line, MSCs were also capable of modulating T cell polarization in a significant proportion of T cell subsets in HC and SSc.

### 3.1. CD4^+^ T Cells

#### 3.1.1. CD4^+^ T Cells from SSc Patients Display an Increased Activation Status Compared to HC

When considering CD4^+^ T cells altogether, a two-fold increase in the percentage of CD25^+^ and HLA-DR^+^ T cells in SSc compared to HC was observed. When dissecting the distinct CD4^+^ T cell subpopulations, the increased cell activation was pronounced in Th1, Th17, Th1/17, CXCR3^−^CCR6^−^ T cells, total Tf helper (Tfh) cells, Tfh-Th1, and Tfh-Th1/17 (*p* < 0.05, Figure 4 and Appendix A). No differences were observed in the percentage of CD4^+^ Treg cells nor in their activation status. Likewise, no differences were found in terms of CD4^+^ T cells’ polarization when comparing SSc and HC (Appendix A).

#### 3.1.2. PHA Stimulation Induces CD4^+^ T Cells’ Activation and Polarization towards Th1, in Both HC and SSc

Upon PHA activation, the percentage of both early- (CD25^+^) and late-activated (HLA-DR^+^) CD4^+^ T cells strongly increased for all cell subsets studied except for Tfh cells, wherein we only verified a statistically significant increase in the percentage of late-activated cells (Figure 5 and Appendix A). Interestingly, PHA induced CD4^+^ T cells to polarize into Th1 while decreasing Th17 compartment for all cell subsets. Of note, a decrease in the CXCR3^−^CCR6^−^ compartment (mainly comprising Th0 and Th2 cells) was observed for CD4^+^ (CXCR5^−^ non-Treg) T cells upon PHA stimulation (Figure 5 and Appendix A).

#### 3.1.3. MSCs Control the Excessive Activation Detected in Unstimulated Th17 and Th1/17 Cells from Systemic Sclerosis Patients and Regulate PHA-Induced Cell Activation in HC and SSc

The effect of MSCs on the activation of unstimulated CD4^+^ T cells varied among the different cell subsets. The presence of MSCs in the cell culture resulted in decreased activation (HLA-DR^+^) of Th1 and Th1/17 in SSc. Conversely, for both SSc and HC, MSCs led to an increased percentage of early-activated (CD25^+^) Tfh cells, which was transversal with respect to all their subsets except Tfh-T1/17. MSCs also induced the activation of Th17-like CD4^+^ follicular regulatory T (Tfr) cells from both SSc and HC, as shown in Figure 5 and Appendix A.

In turn, in the presence of PHA, MSCs decreased cell activation. This effect was transversal to all CD4^+^ T cell subsets (Th1, Th17, Th1/17, CXCR3^−^CCR6^−^, Tfr, CXCR5^−^ Treg, and Tfh), from HC and SSc, and was more pronounced for HLA-DR than CD25 (Figure 5 and Appendix A).

#### 3.1.4. In HC and SSc PBMCs Stimulated with PHA, MSCs Induce Treg, Th17, and Th1/17 Polarization and Suppress PHA-Induced Th1 Differentiation

Concerning CD4^+^ T cell polarization, in general, MSCs induced Th17 and Th1/17 differentiation of unstimulated CD4^+^ T cells in all cell subsets from both the HC and SSc groups. Interestingly, MSCs caused a slight induction or expansion of the CD4+ Treg cells from SSc patients (Figure 5 and Appendix A).

For CD4^+^ T cells stimulated with PHA, MSCs inhibited PHA-induced Th1 polarization and PHA-induced reduction in Th17 and Th1/17 cells; this effect verified in all cell subsets from HC and SSc patients (Figure 5 and Appendix A).

### 3.2. CD8^+^ T Cells

#### 3.2.1. In Systemic Sclerosis, There Is an Increased Percentage of Activated CD8^+^ T Cells, and CD8^+^ Tf Cells Are Preferentially Polarized into Tf-Tc1/Tc17

A three-fold activation increase was observed in the total CD8^+^ T cells from SSc vs. HC. A statistically significant increase in the percentage of activated cells was found among Tc17 and Tc1/17, as illustrated in Figure 6A and Appendix A. A similar tendency was observed for Tc1 and CD8^+^ Tf (*p* > 0.05, Figure 6A). Our data showed a decreased degree of polarization towards Tf-Tc1, which was accompanied by an increase in Tf-Tc1/17 in SSc patients (*p* < 0.05). In the same line, SSc CD8^+^ (CXCR5^−^ non-Treg) T cells tend to by preferentially differentiated into CXCR3^−^CCR6^−^ (a cell compartment comprising Tc2 cells) at the expense of the Tc1 compartment (*p* > 0.05), as detailed in Figure 6B and Appendix A. No differences were found between SSc and HC Treg cells concerning their percentage, activation, or polarization (Appendix A).

#### 3.2.2. PHA Induces CD8^+^ T Cell Activation but Has Little Effect on CD8^+^ T Cell Polarization

In the SSc and HC groups, PHA induced the activation of total CD8^+^ T cells and specific subsets, namely, Tc1, CD8^+^ Treg, total Tf, and Tf-Tc1. In HC, T cell activation was also observed in Tf-CXCR3^−^CCR6^−^ cells. Interestingly, no differences in the percentage of activated cells were detected in SSc Tc17, Tc1/17, Tf-Tc17, and Tf-Tc1/17, which was probably due to the high basal activation level these cells already presented. In general, PHA did not affect CD8^+^ T cells’ polarization; however, we found a decreased percentage of Tf-Tc1 cells and an augmented percentage of CD8^+^CXC5^−^ Treg and Tf-CXCR3^−^CCR6^−^ in both the SSc and HC groups (Figure 7 and Appendix A).

#### 3.2.3. MSCs Suppress Activation of Tc17 Cells from SSc Patients in Both Non-Stimulated and PHA-Stimulated PBMC Cultures

In the non-stimulated PBMCs, MSCs downregulated the increased activation levels found in the Tc17 and CD8^+^CXCR3^−^CCR6^−^ T cells from SSc patients. Of note, a slight increase in Tc17 and Tc1/17 differentiation was found in the HC CD8^+^ T cells co-cultured with MSCs (Figure 7 and Appendix A).

In the presence of PHA, MSCs impaired the PHA-mediated activation of CD8^+^ (CXCR5^−^ non-Treg) T cells, Treg, Tf-Tc1, and Tf-CXCR3^−^CCR6^−^ cells (Figure 7 and Appendix A).

### 3.3. CD4^+^CD8^+^ T Cells

#### 3.3.1. Like CD8^+^ T Cells, Systemic Sclerosis CD4^+^CD8^+^ T Cells Tend to Be More Activated and Polarized into Tf-Tc1/Tc17 Compared to HC

Despite not reaching statistical significance (set at *p* > 0.05), SSc patients exhibited a tendency to present an increased percentage of early-activated cells (CD25^+^) among T17-like, T1/17-like, CXCR3^−^CCR6^−^, and CD4^+^CD8^+^ Tf cells. Like CD8^+^ T cells, CD4^+^CD8^+^ T cells’ polarization tended to be biased towards CXCR3^−^CCR6^−^, whereas the T1-like subset was depleted (*p* > 0.05); moreover, the percentage of CD4^+^CD8^+^ Tf-T1/17 cells was increased in SSc patients and, conversely, that of CD4^+^CD8^+^ Tf-T1 cells was decreased (*p* > 0.05). No differences were found concerning Treg percentage, activation, and polarization. Detailed data on CD4^+^CD8^+^ T cell subsets for SSc vs. HC groups can be found in Appendix A.

#### 3.3.2. PHA Induces CD4^+^CD8^+^ T Cells’ Activation, with No Effect on Their Polarization

PHA stimulation resulted in an increased percentage of early-(CD25^+^) and late-activated (HLA-DR^+^) CD4^+^CD8^+^ T cells when considering this cell population altogether as well as in T1-like, T1/17-like, and CXCR3^−^CCR6^−^ (CXCR5^−^ non-Treg) CD4^+^CD8^+^ T cells and CD4^+^CD8^+^ Tf cells from HC and SSc patients, as shown in Figure 8 and Appendix A. On the other hand, PHA did not affect CD4^+^CD8^+^ T cell polarization under our experimental conditions, although a tendency toward a decrease in T1-like cells under the influence of PHA (*p* > 0.05) in HC and SSc patients was detected (Appendix A).

#### 3.3.3. MSCs Induce CD4^+^CD8^+^ Treg Cells and Suppress PHA-Induced Activation

CD4^+^CD8^+^ Treg cells were induced and/or expanded in the presence of MSCs. A similar effect was observed in CD4^+^CD8^+^ T17- and T1/17-like cells. MSCs also had the ability to suppress PHA-induced CD4^+^CD8^+^ T cell activation (Figure 8 and Appendix A).

### 3.4. CD4^−^CD8^−^TCRαβ T Cells

#### 3.4.1. Systemic Sclerosis Patients Exhibit a Significant Increase in CXCR3^−^CCR6^−^CD4^−^CD8^−^TCRαβ T Cells, and Tf Cells Are Preferentially Polarized towards T1/17

SSc patients presented a tendency to present an increased percentage of late-activated (HLA-DR^+^) CD4^−^CD8^−^ TCRαβ T cells (*p* > 0.05), which was observed in all the cell subpopulations analyzed (Appendix A). As in the CD8^+^ T cells, there was an increase in the CXCR3^−^CCR6^−^ cell compartment, which occurred at the expense of T1-like cells (*p* > 0.05), and an augmentation of Tf-T17 (*p* < 0.05) and Tf-T1/17-like cells (*p* > 0.05), which was accompanied by a reduction in the Tf-T1 cell compartment (*p* > 0.05), in SSc CD4^−^CD8^−^ TCRαβ T cells (Figure 9 and Appendix A). No differences between SSc and HC were verified for CD4^−^CD8^−^ TCRαβ Treg cells (Appendix A).

#### 3.4.2. PHA Activates CD4^−^CD8^−^ TCRαβ T Cells Isolated from HC and SSC Patients and Affects Their Polarization in HC

Upon PHA stimulation, increased percentages of CD25^+^ and HLA-DR^+^ cells were found in T1-like and T1/17-like (CXCR5^−^ non-Treg) CD4^−^CD8^−^ TCRαβ T cells as well as in total Tf and Tf-T1-like CD4^−^CD8^−^ TCRαβ T cells, which were more pronounced in the HC group than in the SSc group. PHA did not affect the polarization of CD4^−^CD8^−^ TCRαβ T cells from SSc patients but induced a decrease in T1-like cells and an increase in CXCR3^−^CCR6^−^ cells from HC patients, as shown in Figure 10 and Appendix A.

#### 3.4.3. MSCs Regulate Both the Activation and Polarization of CD4^−^CD8^−^TCRαβ T Cells

In the presence of MSCs, the HC-unstimulated T cells polarized into T17-like cells and, at the same time, their activation was impaired (*p* < 0.05). A similar trend was observed for T1/17 cells (*p* > 0.05). In turn, T1-like cell compartment decreased in the presence of MSCs, but their activation status was not affected. In the presence of PHA, MSCs impaired PHA-mediated CD4^−^CD8^−^ TCRαβ T cell activation (Figure 10 and Appendix A).

### 3.5. γδ. T Cells

#### 3.5.1. γδ. T Cells from Systemic Sclerosis Patients Display an Increased Activation Status and a Polarization Bias toward T1/17 and CXCR3^−^CCR6^−^Cell Compartments

In the SSc samples, an increased percentage of late-activated (HLA-DR^+^) γδ T cells was detected in all the cell subsets analyzed, reaching statistical significance for the T1/17, CXCR3^−^CCR6^−^, and Tf compartments (Figure 11A and Appendix A). In parallel with CD8^+^ T cells, γδ T cells showed an increase in γδ T1/17-like, CXCR3^−^CCR6^−^ cells, and Tf-T1/17, which was accompanied by a depletion in γδ T1-like cells (*p* < 0.05) (Figure 11B and Appendix A). No difference was found concerning the γδ Treg cell percentage between the SSc and HC groups (Appendix A).

#### 3.5.2. PHA Induces Activation and Modulates Polarization of γδ T Cells in HC and SSc

Upon PHA stimulation, cell activation occurred in all the HC and SSc γδ T cell subsets analyzed except in those γδ T cell subsets from the SSc patients for which a great degree of basal activation was detected, namely, T1/17, CXCR3^−^CCR6^−^, and Tf γδ T cells. A decreased polarization toward T1 and an increased percentage of CXCR3^−^CCR6^−^ γδ T cells were also observed (Figure 12 and Appendix A). γδ Treg subsets were not analyzed due to the low number of cells obtained.

#### 3.5.3. MSCs Inhibit the Strongly Activated Tf γδ T Cells Isolated from SSc Patients

When MSCs were cultured with unstimulated PBMCs, a bias towards the T1/17 phenotype was observed, which was accompanied by the decreased activation of CXCR3^−^CCR6^−^ γδ T cells. Importantly, MSCs inhibited the basal activation verified in the (unstimulated) γδ T cells from the SSc samples, which displayed a high activation profile. MSCs also downregulated the cellular activation induced by PHA. No changes were detected in the percentage of γδ Treg cells (Figure 12 and Appendix A).

### 3.6. Comparison between ACA and Anti-Scl-70 SSc Patients

Despite the small number of SSc patients and the limited conclusions that could be drawn after their subdivision into ACA (*n* = 4) and anti-Scl-70 (*n* = 5) SSc groups, relevant differences were found between these two groups. An increased level of basal activation was found in both SSc groups (vs. HC), which was significantly higher in the Th17 and CD4^+^ CXCR3^−^CCR6^−^ T cells of the ACA vs. anti-Scl-70 SSc patients (Table 4). The CD8^+^ T cells isolated from the anti-Scl-70 SSc samples were preferentially differentiated towards a CXCR3^−^CCR6^−^ phenotype, which was accompanied by a decrease in the Tc1 compartment, when compared to ACA SSc, which displayed a distribution similar to that of HC (Table 4). Interestingly, an increased percentage of CD4+CD8+ Treg cells was found in anti-Scl-70 vs. ACA SSc; in turn, the percentage of these cells in ACA SSc patients was reduced compared to the HC (Table 4).

Interestingly, under the influence of MSCs, the CD4^+^ T cells from the anti-Scl-70 SSc group were more prone to undergoing differentiation into Th17 (vs. HC and ACA SSc), while in the ACA SSc group, the presence of MSCs resulted in a decrease in Th1/17, Tc1, and Tc1/17 differentiation, while CD4^+^ Treg cells were induced and/or expanded (Appendix A). Though it was observed that MSCs possess the ability to induce/expand CD8^+^ Treg and CD4^+^CD8^+^ Treg cells in HC and ACA SSc patients, in the anti-Scl-70 SSc patients, the process Treg induction/expansion mediated by MSCs seemed to be less efficient. In addition, it was observed that MSCs could increase the expression of the activation marker CD25 in the Tc1/17 cells from anti-Scl-17 patients but not in the ACA SSc group (Appendix A). Finally, an effective inhibition of T cell late-activation mediated by MSCs was achieved in both the ACA and anti-SCl-70 SSc patients (Appendix A).

## 4. Discussion

In the last two decades, the refinement of flow cytometry technology has provided the possibility of evaluating the expression of different proteins simultaneously; consequently, a multitude of T cell subsets has been described. This study focused on obtaining a broad understanding on the immunomodulatory effect of MSCs on each different T cell subpopulation. Accordingly, we performed a detailed identification of T cell subsets, yielding 64 subsets, and analyzed how MSCs modulated their activation and polarization. This approach allowed us to evaluate the effect of MSCs on the numerous T cell subsets involved in SSc pathogenesis and describe, for the first time, the effect of MSCs in minor T cell subsets.

Given the heterogeneity of SSc, efforts have been made to refine its classification and determine the adequate criteria with which to subclassify SSc patients into subgroups with prognostic value. A patient’s autoantibody profile is a valuable criterion as it is associated with clinical features and prognosis. ACAs are associated with limited skin involvement, isolated pulmonary arterial hypertension (without interstitial lung disease), digital ulcers and gangrene, and calcinosis [51,52]. Of note, ACA SSc patients display a decreased risk of scleroderma renal crisis [53]. In turn, anti-Scl-70 is associated with diffuse cutaneous involvement, interstitial lung disease, digital ulcers and gangrene, severe heart disease, and scleroderma renal crisis with a poor outcome [1,51,52,53]. Interestingly, there is a strong association between anti-Scl-70 antibodies and human leukocyte antigen (HLA) alleles, thereby establishing a link between T cells and the autoantibody profile [54]. Thus, although this study included a small cohort of patients, which limited the conclusions that could be drawn, SSc patients were subdivided into ACA and anti-Scl-70 subgroups. The results obtained support the notion that MSCs exert an efficient immunomodulatory effect on T cells from ACA and anti-Scl-70 SSc patients. In addition, immune modulation was transversal with respect to the different T cell subsets studied for both SSc groups.

The roles of Th1 and Th17 in SSc pathogenesis became evident when high levels of CXCL9, CXCL10 (CXCR3 ligands) [55], and CCL20 (CCR6 ligand) [56] were detected in SSc skin lesions, indicating an active migration of T cells bearing these receptors (Th1, Th17, and Th1/17 cells) to these sites. Later, it was demonstrated that CXCR3 plays a role in T cell migration into the inflamed skin [57]. More recently, CCR6 was shown to mediate Treg migration into the skin of vitiligo patients [58]. The percentage of Th1 in SSc PB was found to be decreased [59,60,61] or unchanged [62] and accompanied by a bias toward Th2 polarization [1,61,62], but Th1 cells from SSc patients present an activated phenotype [63], indicating that they play an active role in SSc. Likewise, increased percentages of circulating Th17 [21,23,59,60,62,64,65,66,67] and IL-17 serum levels [65,68,69] were found in SSc patients, which were even more pronounced in dcSSc [65]. Th17 percentage has been shown to be positively correlated with disease duration [64] and interstitial lung disease [62]. Accordingly, Th17-rich skin [21,66,69,70,71] and lung [21] infiltrate are detected in SSc patients, and there is a positive correlation between the Th17 cells in the infiltrate and the severity of a skin lesion [70]. Increased levels of Th17 cell activation are also found in SSc patients [63]. Several studies have demonstrated that IL-17A possesses pro-fibrotic properties in both human and SSc animal models [17,21,65,66,69,71,72]; however, there are some contradictory findings in humans [66,71]. Notwithstanding, IL-17A, IL-17F, and IL-17E were demonstrated to promote vasculopathy in SSc patients [65,67,69]. A recent clinical trial assessing the safety and efficacy of brodalumab (a fully human anti–IL-17 receptor A monoclonal antibody) was carried out using eight SSc patients; the trial demonstrated its safety and that the blockage of the IL-17 signaling pathway resulted in the improvement of skin fibrosis, a decrease of the mRSS, a reduction in digital ulcers, and increased Treg/Th17 ratios in SSc patients [73].

As with CD4^+^ T cells, the percentage of Tc1 in SSc PB was found to be decreased and accompanied by a bias toward Tc2 polarization [1,61,62]. Importantly, circulating CD8^+^ T cells that produce IL-13 express skin-homing receptors. These cells accumulate in SSc skin lesions and exert cytotoxic activity against endothelial cells and pro-fibrotic activity mediated by IL-13 and IL-4 secretion [17,74,75,76,77].

Accordingly, our results show an increased degree of basal activation of T cells from SSc, namely, Th17, Th1/17, Tc17, and Tc1/17, indicating the active role of these cells in SSc. In addition, our study demonstrates that MSCs suppress the activation of all these cell subsets. Though, according to our results MSCs promote polarization towards Th1/17, which could be an adverse effect for the treatment of SSc, the MSC-induced Th1/17 cells display a low activation profile.

There are controversial results concerning alterations in the percentage of Treg cells, with some studies describing an increased percentage of CD4^+^ Treg in SSc patients [64,78,79,80], for which there is a positive correlation with disease severity [79] and interstitial lung disease [64], and others showing decreased or unchanged Treg levels [23,59,60,66,67]. Nonetheless, all the studies published so far indicate that both CD4^+^ and CD8^+^ Tregs from SSc patients are dysfunctional [15,20,59,65,67,80]. They fail to inhibit T cell proliferation [59,78] and produce lower levels of TGF-β and IL-10 [15,20,65,67,78,80]. IL-10 exerts anti-fibrotic activity and inhibits collagen production by fibroblasts [81]. SSc skin lesions present a reduction in Treg infiltration, and Treg-infiltrating cells have impaired regulatory ability [66]. Of note, in SSc, both circulating and skin-infiltrated Treg cells acquire a Th2-like phenotype, producing IL-4 and IL-13 and thus promoting skin fibrosis [82].

We found no changes in Treg percentage or activation between the SSc and HC groups. Remarkably, under our experimental conditions, MSCs increased the percentage of CD4^+^ and CD4^+^CD8^+^ Treg cells in SSc without changing their activation profile. Interestingly, MSCs also biased CD4^+^ Tfr cells toward a Th17- and Th1/17-like phenotype.

As autoantibodies are a hallmark of SSc, the role of Tfh cells is ineluctable. Tfh cells support B cell survival, differentiation into plasma cells, antibody production, and immunoglobulin class-switching. Increased levels of circulating Tfh were described in SSc PB [21], particularly in early diffuse cutaneous systemic sclerosis [23], though controversial results have been published [22,68]. Tfh count correlates with disease severity, including skin and vascular lesions, and is negatively correlated with CD24^hi^CD27^+^ Breg cells [63,83,84,85]. Tfh cells display an activated phenotype [22,63,83] and infiltrate SSc skin lesions [68,84]. Besides their essential role in antibody production, SSc Tfh cells exert pro-fibrotic activity mediated by IL-21, which induces fibroblastsè transdifferentiation into myofibroblasts [17,22,83,84]. Accordingly, Tfh cells from SSc patients secrete higher amounts of IL-21 and IL-17 and present increased percentages of Tfh1 and Tfh17 cells [68,83].

Herein, we report increased activation levels of Tfh1 and Tfh1/17, CD8^+^ Tf, and γδ Tf cells from SSc. Interestingly, MSCs were able to inhibit Tfh1/17 and γδ Tf basal activation but also promoted the further activation of Tfh1 cells. Importantly, Tfh17 cells have been linked to autoimmunity and the production of autoantibodies [86]. In the same line, SSc patients display an increased percentage of Tc1/17-like (CD8^+^) Tfc cells as well as CD4^−^CD8^−^ Tf-T17 and Tf-T1/17 cells. Tc1/17-like (CD8^+^) Tfc can assume different functions depending on the environment, including cytotoxic, supportive, and regulatory activities [87].

The few studies evaluating CD4^+^CD8^+^ T cells in SSc reported increased percentages of these cells in the PB and fibrotic skin. Notably, SSc skin-infiltrating CD4^+^CD8^+^ T cells display cytotoxic activity and produce high levels of IL-4 [17,77]. We found no statistically significant differences between SSc and HC for this cell population, but there was a tendency toward the increased activation of CD4^+^CD8^+^ T in SSc as well as CD4^−^CD8^−^ T and γδ T cells. According to our study, MSCs induce/expand CD4^+^CD8^+^ Treg cells from SSc and HC patients and regulate the activation of CD4^+^CD8^+^ T, CD4^−^CD8^−^ T, and γδ T cells from SSc patients. MSCs also induces T17-like cells in CD4^+^CD8^+^ and CD4^−^CD8^−^ T cells and T1/17-like cells in CD4^+^CD8^+^, CD4^−^CD8^−^, and γδ T cells. Several studies have indicated the relevance of γδ T cells in SSc. These cells accumulate in the skin and lungs of SSc patients and display higher cytotoxic activity compared to their healthy counterparts [88]. When studied with higher detail, the Vγ9^+^ [89] and δ1^+^ [90] γδ T cell subsets were found to be expanded in the PB and skin from SSc and display an activated phenotype [90,91,92]. PB Vγ9^+^ γδ T cells showed increased granzyme B expression [89]. Moreover, in vitro cell culture assays demonstrated that PB γδ T cells from SSc patients induced a higher fibroblast proliferation rate and increased collagen synthesis compared to γδ T cells from healthy individuals [91,93]. Thus, the expanded subsets of γδ T cells found in SSc patients seem to be endowed with stronger cytotoxic and pro-fibrotic activity, thus contributing to the tissue fibrosis that characterizes this pathology. CCR6 marker identifies γδ T17 cells, which constitute a cell subset with skin- and mucosa-homing properties that can be induced extra-thymically upon skin inflammation. This cell subset is found in the dermis and plays an important role in the immunity of the epithelial surfaces [94,95]. It plays an active role in auto-immune diseases, such as ankylosing spondylitis and multiple sclerosis [94,95], but its role in SSc is yet to be unveiled. More studies focusing on the function of minor T cell subsets in healthy individuals and SSc patients are needed to broadly understand how the MSC-mediated immunomodulation of these cells can impact SSc.

## 5. Conclusions

This study demonstrated that MSCs’ ability to inhibit T cell activation was transversal with respect to several T cell subsets for both HC and SSc and was more pronounced for the later stages of T cell activation. Notably, SSc patients displayed an increased activation status in some T cell subsets, all of which were downregulated by MSCs. MSCs also biased T cell polarization towards Th17/T17-like non-activated cells at the expense of Th1/T1-like and CXCR3^−^CCR6^−^ cell compartments.

Although the small number of patients included in this research is a limitation and, therefore, in vitro studies enrolling larger cohorts of SSc patients are recommended to support the translation of our findings to clinical studies, our research provides a deeper understanding of the effect of MSCs on T cell subsets that actively participate in SS as well as the effect on minor T cell subsets that are poorly studied. The transversal inhibition of activation among several T cell subsets, including those implicated in SSc pathogenesis, and the ability to modulate T cell polarization in vitro supports the potential of using MSC-based therapies to regulate T cells in a disease whose onset and development may be due to the malfunction of the immune system. Finally, although we are aware of all the limitations and challenges that translation from in vitro to in vivo models and clinical trials imply, we find that these results validate the potential use of iATMP SLCTmsc02 as an innovative therapy for SSc patients and justify the further pre-clinical and clinical development of this MSC-based product.

## Figures and Tables

**Figure 1 biomedicines-11-01329-f001:**
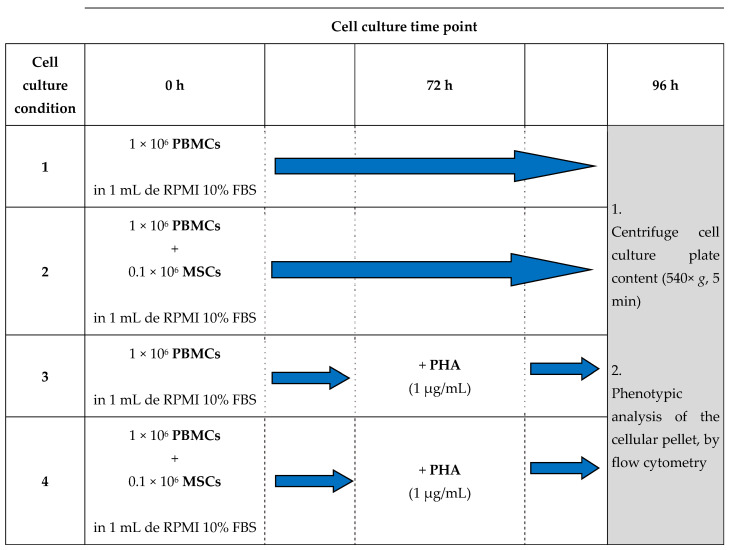
Cell culture conditions. PBMCs from SSc patients and HC were cultured for 96 h in the presence/absence of MSCs and in the presence/absence of PHA in the last 24 h of cell culture, thereby establishing four different cell culture conditions. FBS, fetal bovine serum; MSCs, mesenchymal stromal cells; PBMCs, peripheral blood mononuclear cells; PHA, phytohemagglutinin.

**Figure 2 biomedicines-11-01329-f002:**
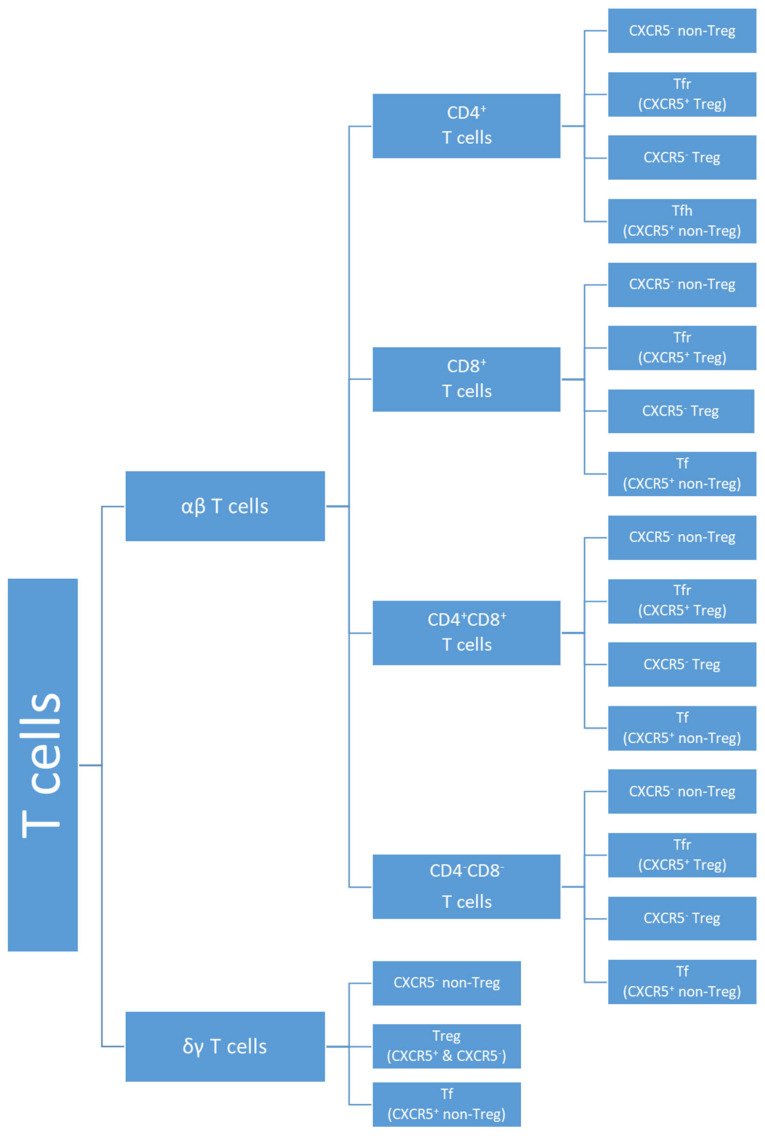
Scheme representing the T cell subsets identified.

**Figure 3 biomedicines-11-01329-f003:**
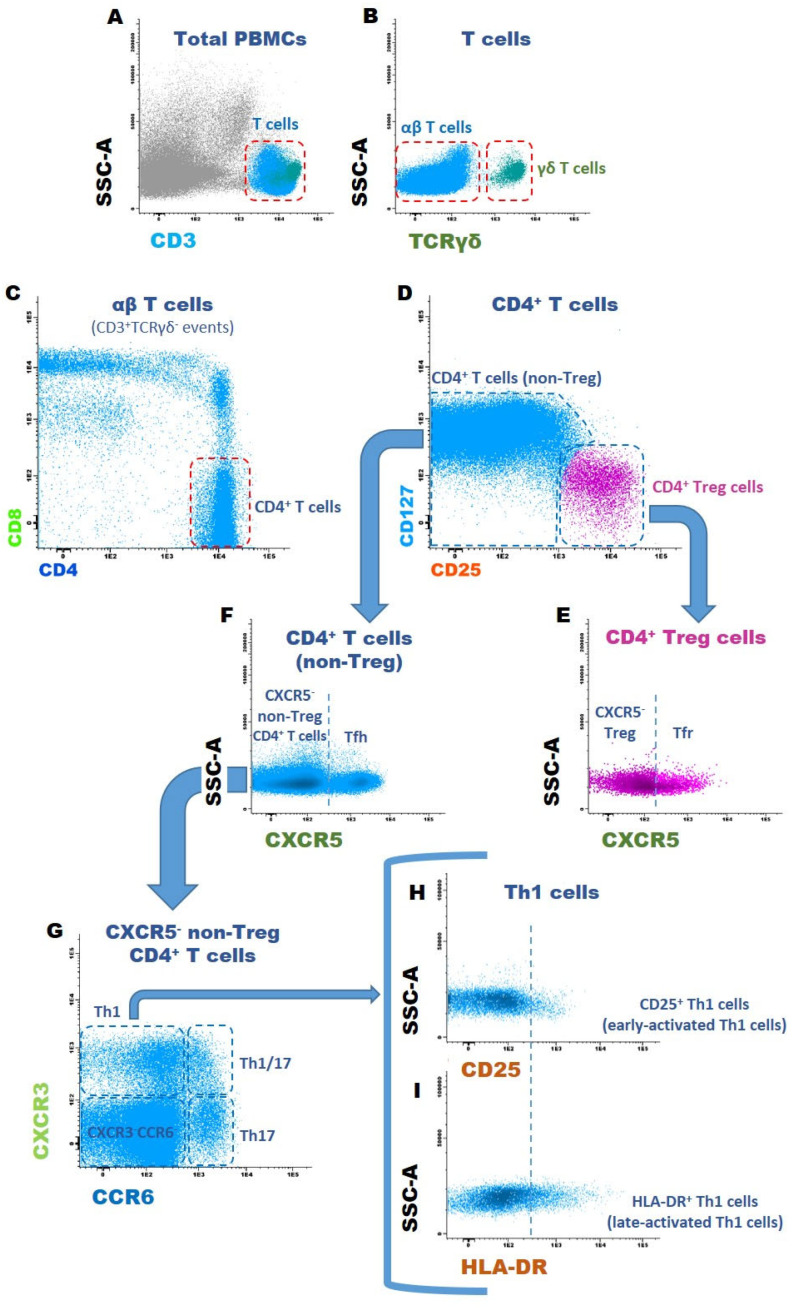
Gating strategy used to identify T cell subsets. Dot plot histograms illustrating the identification of 16 cell subsets within CD4^+^ T cells, and the strategy for evaluating the percentage of early- and late-activated cells. The same approach was applied to CD8^+^, CD4^+^CD8^+^, CD4^−^CD8^−^ αβ T cells, and δγ T cells, allowing for the identification of 80 cell T subsets. (**A**) T cells were identified based on CD3 expression and SSC light dispersion properties. (**B**) Identification of αβ T cells and γδ T cells according to TCRγδ expression. (**C**) Identification of CD4^+^, CD8^+^, CD4^+^CD8^+^, and CD4^−^CD8^−^ T cells within αβ T cells. (**D**) CD25 vs. CD127 dot plot for distinguishing CD4^+^ Treg cells from CD4^+^ non-Treg cells. (**E**) CD4^+^ Tregs were subdivided into follicular regulatory (CXCR5^+^) T cells (Tfr) and (CXCR5^−^) Tregs, and (**F**), within non-Treg cells, the CD4^+^ follicular T cells were identified as CXCR5-positive events; CXCR5^−^ non-Tregs were spotted in the same dot plot. (**G**) CXCR3 vs. CCR6 dot plot illustrating how (CXCR5^−^ non-Treg) CD4^+^ T cells were subdivided into Th1, Th17, Th1/17, and CXCR3^−^CCR6^−^ Th cells. The same approach was used for all T cell subsets in order to identify T1-, T17-, T1/17-like, and CXCR3^−^CCR6^−^ T cells. (**H**) Dot plot histograms displaying the activation markers CD25 and (**I**) HLA-DR; the percentages of early- (CD25^+^) and late-activated (HLA-DR^+^) T cells were determined as depicted in the histogram. SSC-A, side scatter light dispersion.

**Figure 4 biomedicines-11-01329-f004:**
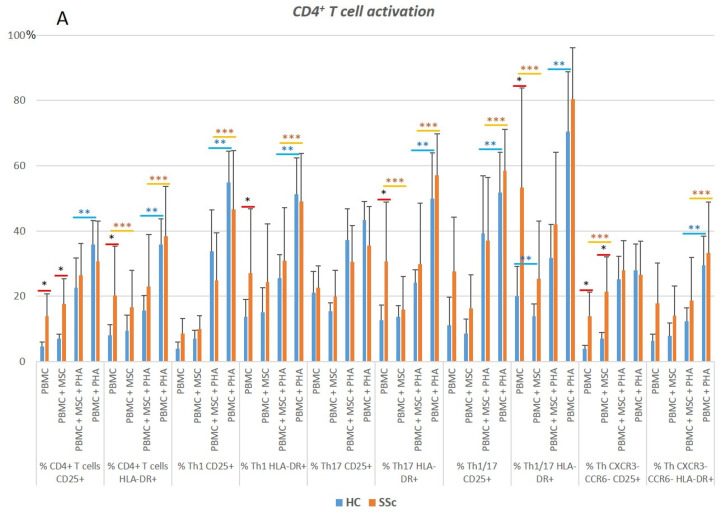
CD4^+^ T cell activation status. Activation status of (**A**) CD4^+^ T cells and (**B**) CD4^+^ Tfh cells in HC and SSc at basal level (PBMC) and effect of the different culture conditions on the activation of T cell subsets. * HC vs. SSc in the same culture conditions. ** Comparison between different culture conditions within the HC group: unstimulated PBMCs (PBMC) vs. PBMCs co-cultured with MSCs (PBMC + MSC), and PHA-stimulated PBMCs (PBMC + PHA) vs. PBMCs co-cultured with MSCs and stimulated with PHA (PBMC + MSC + PHA). *** Comparison between different culture conditions within the SSc group: PBMC vs. PBMC+MSC, and PBMC+PHA vs. PBMC + MSC + PHA. Mann–Whitney, Friedman, and Wilcoxon tests were applied when appropriate. Differences were considered statistically significant when *p* < 0.05. HC, healthy controls; MSCs, mesenchymal stromal cells; PBMCs, peripheral blood mononuclear cells; PHA, phytohemagglutinin; SSc, systemic sclerosis.

**Figure 5 biomedicines-11-01329-f005:**
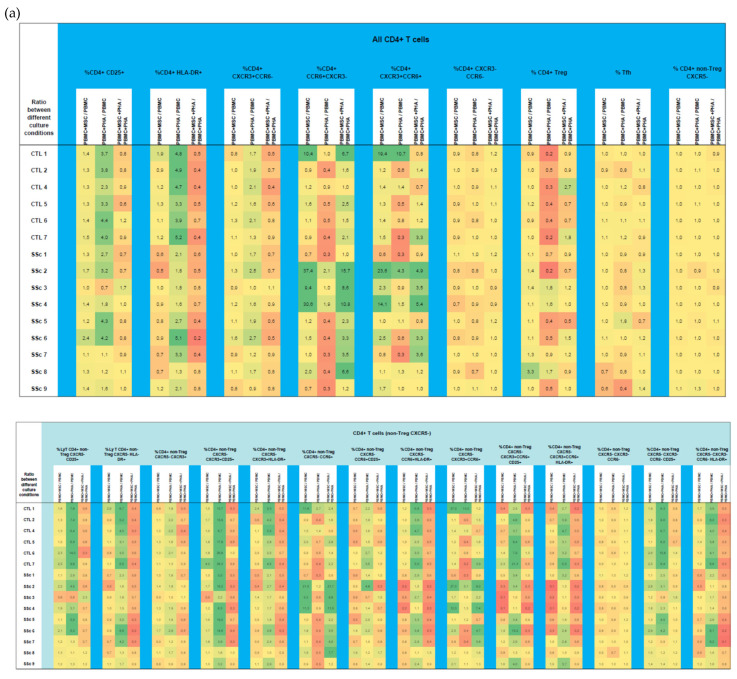
Effect of MSCs’ and PHA stimulation on the polarization and activation of CD4^+^ T cells isolated from HC and SSc patients. MSCs effect was analyzed on CD4^+^ T cells as a whole (a), and within each one of the subsets: CXCR5^−^ non-Treg (**a**), Treg (**b**), and follicular T cells (**c**). Results are presented as a ratio between PBMC + MSC/PBMC (indicating the effect of MSCs on unstimulated PBMCs), PBMC + PHA/PBMC (to assess the effect of PHA on PBMCs), and PBMC+MSC+PHA/PBMC + PHA (to evaluate the effect of MSCs on PHA-stimulated PBMCs) for each participant. Yellow denotes no effect or minimal effect (ratio ≈ 1), orange to red indicates inhibition (ratio ≤ 0.7), and green represents induction/stimulation (ratio ≥ 1.5). HC, healthy controls; MSCs, mesenchymal stromal cells; PBMCs, peripheral blood mononuclear cells; PHA, phytohemagglutinin; SSc, systemic sclerosis.

**Figure 6 biomedicines-11-01329-f006:**
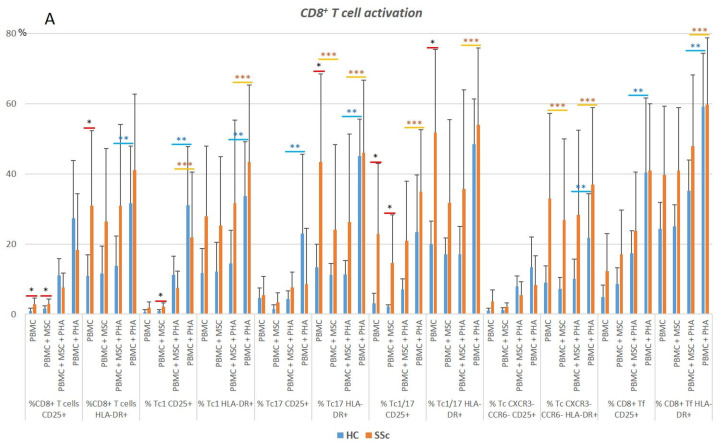
Characterization of CD8^+^ T cell activation status and polarization. (**A**) Activation status of CD8^+^ T cells in HC and SSc at basal level (PBMC) and effect of the different culture conditions on the activation of CD8^+^ T cell subsets. (**B**) Distribution of CD8^+^ T cells among the distinct differentiated subsets, and influence of the culture conditions on CD8^+^ T cells polarization. * HC vs. SSc in the same culture conditions. ** Comparison between different culture conditions within the HC group: unstimulated PBMCs (PBMC) vs. PBMCs co-cultured with MSCs (PBMC + MSC) and PHA-stimulated PBMCs (PBMC + PHA) vs. PBMCs co-cultured with MSCs and stimulated with PHA (PBMC + MSC + PHA). *** Comparison between different culture conditions within the SSc group: PBMC vs. PBMC + MSC and PBMC + PHA vs. PBMC + MSC + PHA. Mann–Whitney, Friedman, and Wilcoxon tests were applied when appropriate. Differences were considered statistically significant at *p* < 0.05. HC, healthy controls; MSCs, mesenchymal stromal cells; PBMCs, peripheral blood mononuclear cells; PHA, phytohemagglutinin; SSc, systemic sclerosis.

**Figure 7 biomedicines-11-01329-f007:**
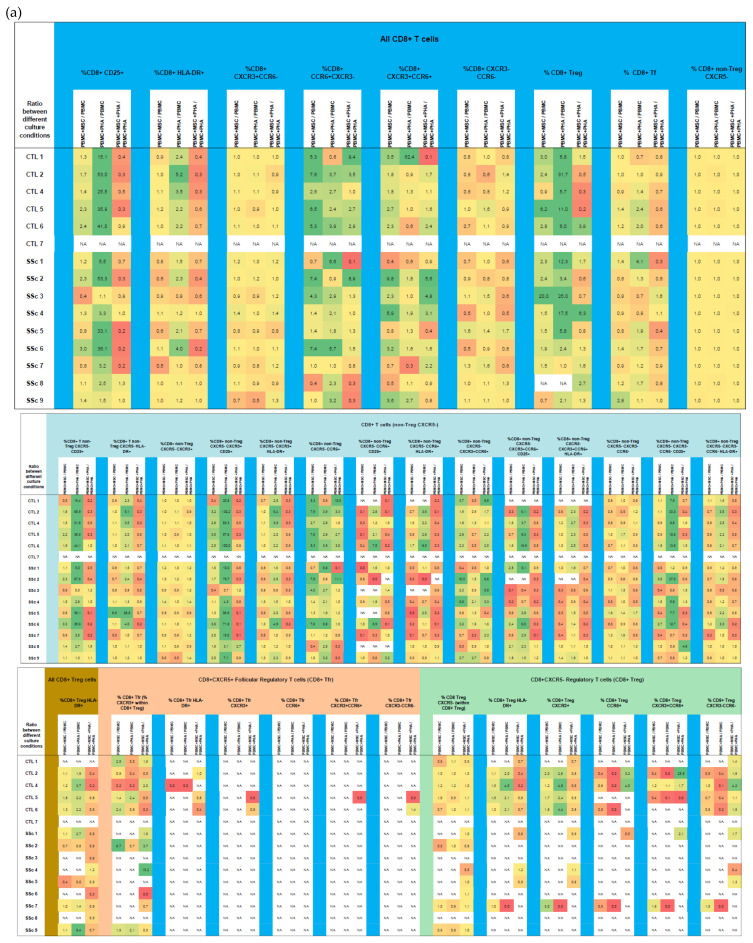
Effect of MSCs’ and PHA stimulation on the polarization and activation of CD8^+^ T cells from HC and SSc patients. MSCs effect was analyzed on CD8^+^ T cells as a whole (**a**), and within each one of the subsets: CXCR5^−^ non-Treg (**a**), Treg (**a**), and follicular T cells (**b**). Results are presented as a ratio between PBMC + MSC/PBMC (indicating the effect of MSCs on unstimulated PBMCs), PBMC + PHA/PBMC (to assess the effect of PHA on PBMCs), and PBMC + MSC + PHA/PBMC + PHA (to evaluate the effect of MSCs on PHA-stimulated PBMCs) for each participant. Yellow denotes no effect or minimal effect (ratio ≈ 1), orange to red indicates inhibition (ratio ≤ 0.7), and green represents induction/stimulation (ratio ≥ 1.5). HC, healthy controls; MSCs, mesenchymal stromal cells; PBMCs, peripheral blood mononuclear cells; PHA, phytohemagglutinin; SSc, systemic sclerosis.

**Figure 8 biomedicines-11-01329-f008:**
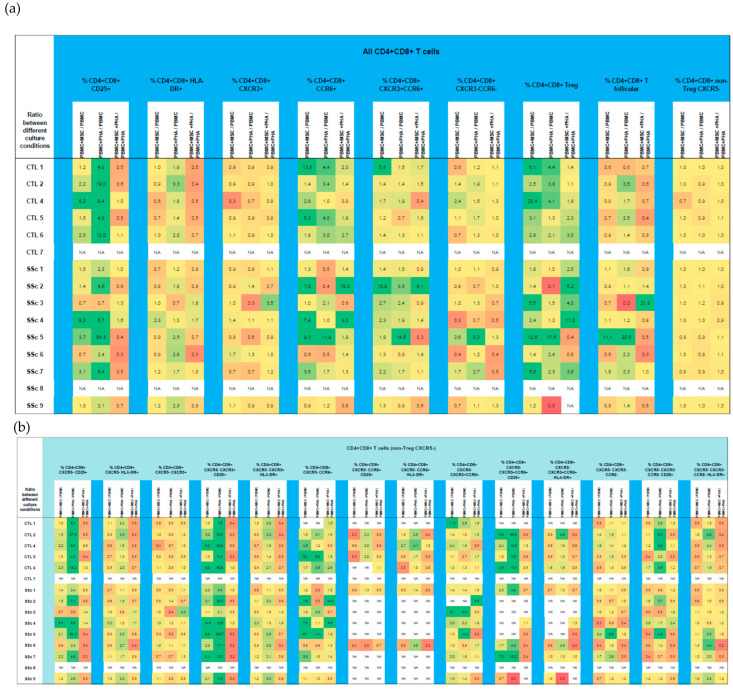
Effect of MSCs’ and PHA stimulation on the polarization and activation of CD4^+^CD8^+^ T cells isolated from HC and SSc patients. MSCs effect was analyzed on CD4^+^ T cells as a whole (**a**), and within each one of the subsets: CXCR5^−^ non-Treg (**b**), Treg (**c**), and follicular T cells (**c**). Results are presented as a ratio between PBMC + MSC/PBMC (indicating the effect of MSCs on unstimulated PBMCs), PBMC+PHA/PBMC (to assess the effect of PHA on PBMCs), and PBMC + MSC + PHA/PBMC + PHA (to evaluate the effect of MSCs on PHA-stimulated PBMCs) for each participant. Yellow denotes no effect or minimal effect (ratio ≈ 1), orange to red indicates inhibition (ratio ≤ 0.7), and green represents induction/stimulation (ratio ≥ 1.5). HC, healthy controls; MSCs, mesenchymal stromal cells; PBMCs, peripheral blood mononuclear cells; PHA, phytohemagglutinin; SSc, systemic sclerosis.

**Figure 9 biomedicines-11-01329-f009:**
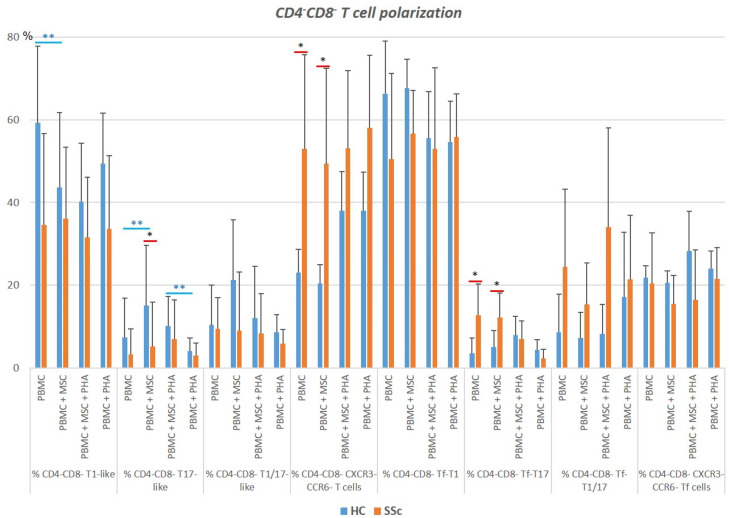
Characterization of CD4^−^CD8^−^ TCRαβ T cell polarization. Distribution of CD4^−^CD8^−^ TCRαβ T cells isolated from HC and SSc patients among the differentiated subsets, and influence of the culture conditions on their polarization. * HC vs. SSc in the same culture conditions. ** Comparison between different culture conditions within the HC group: unstimulated PBMCs (PBMC) vs. PBMCs co-cultured with MSCs (PBMC + MSC), and PHA-stimulated PBMCs (PBMC+PHA) vs. PBMCs co-cultured with MSCs and stimulated with PHA (PBMC + MSC + PHA). Mann–Whitney, Friedman, and Wilcoxon tests were applied when appropriate. Differences were considered statistically significant at *p* < 0.05. HC, healthy controls; MSCs, mesenchymal stromal cells; PBMCs, peripheral blood mononuclear cells; PHA, phytohemagglutinin; SSc, systemic sclerosis.

**Figure 10 biomedicines-11-01329-f010:**
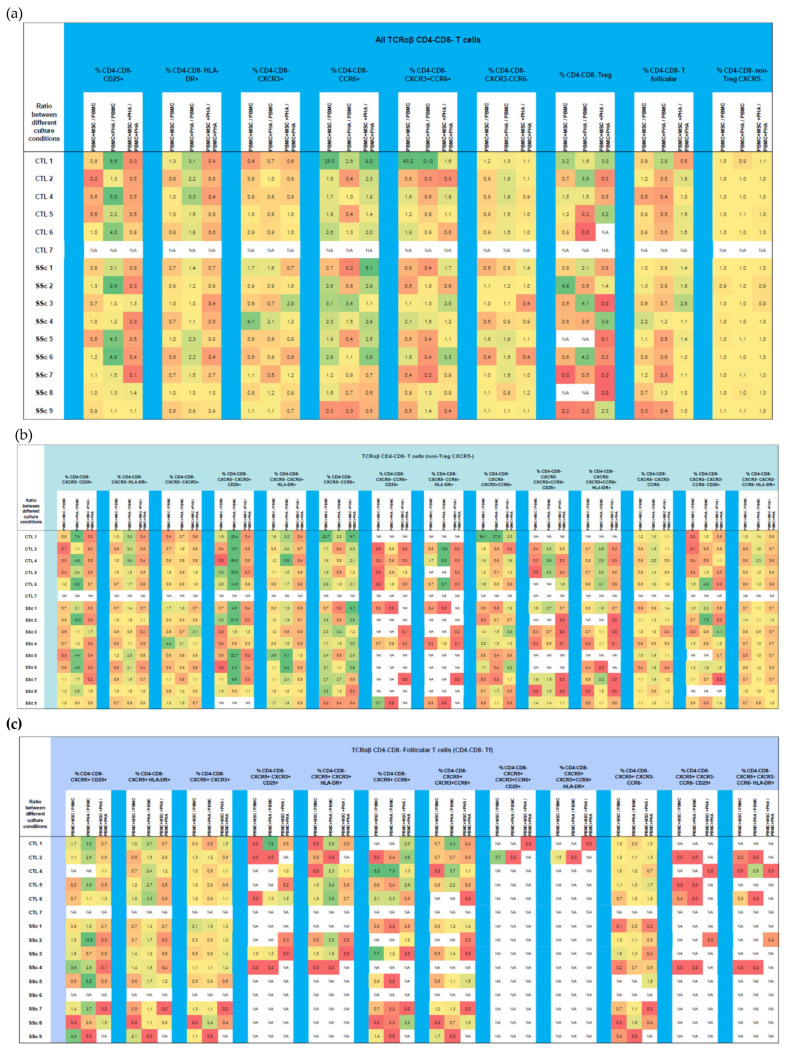
Effect of MSCs’ and PHA stimulation on the polarization and activation of CD4^−^CD8^−^ TCRαβ T cells isolated from HC and SSc patients. MSCs effect was analyzed on CD4^−^CD8^−^ T cells as a whole (**a**), and within each one of the subsets: CXCR5^−^ non-Treg (**b**), and follicular T cells (**c**). Results are presented as a ratio between PBMC + MSC/PBMC (indicating the effect of MSCs on unstimulated PBMCs), PBMC+PHA/PBMC (to assess the effect of PHA on PBMCs), and PBMC + MSC + PHA/PBMC + PHA (to evaluate the effect of MSCs on PHA-stimulated PBMCs) for each participant. Yellow denotes no effect or minimal effect (ratio ≈ 1), orange to red indicates inhibition (ratio ≤ 0.7), and green represents induction/stimulation (ratio ≥ 1.5). HC, healthy controls; MSCs, mesenchymal stromal cells; PBMCs, peripheral blood mononuclear cells; PHA, phytohemagglutinin; SSc, systemic sclerosis.

**Figure 11 biomedicines-11-01329-f011:**
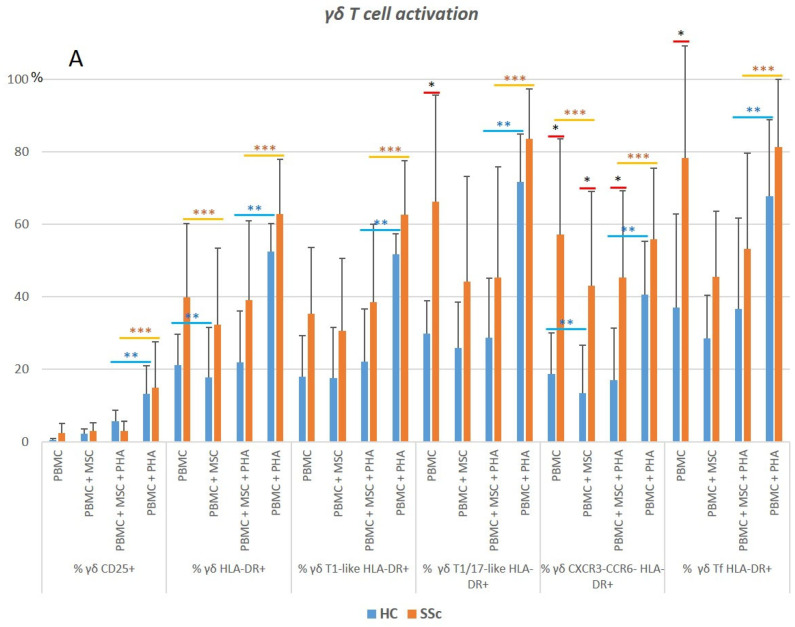
Characterization of γδ T cell activation status and polarization. (**A**) Activation status of γδ T cells in HC and SSc patients at basal level (PBMC), and the effect of the different culture conditions on the activation of γδ T cell subsets. (**B**) Distribution of γδ T cells among the distinct subsets, and influence of the culture conditions on γδ T cells’ polarization. * HC vs. SSc in the same culture conditions. ** Comparison between different culture conditions within the HC group: unstimulated PBMCs (PBMC) vs. PBMCs co-cultured with MSCs (PBMC + MSC), and PHA-stimulated PBMCs (PBMC + PHA) vs. PBMCs co-cultured with MSCs and stimulated with PHA (PBMC + MSC + PHA). *** Comparison between different culture conditions within the SSc group: PBMC vs. PBMC + MSC, and PBMC+PHA vs. PBMC+MSC+PHA. Mann–Whitney, Friedman, and Wilcoxon tests were applied when appropriate. Differences were considered statistically significant at *p* < 0.05. HC, healthy controls; MSCs, mesenchymal stromal cells; PBMCs, peripheral blood mononuclear cells; PHA, phytohemagglutinin; SSc, systemic sclerosis.

**Figure 12 biomedicines-11-01329-f012:**
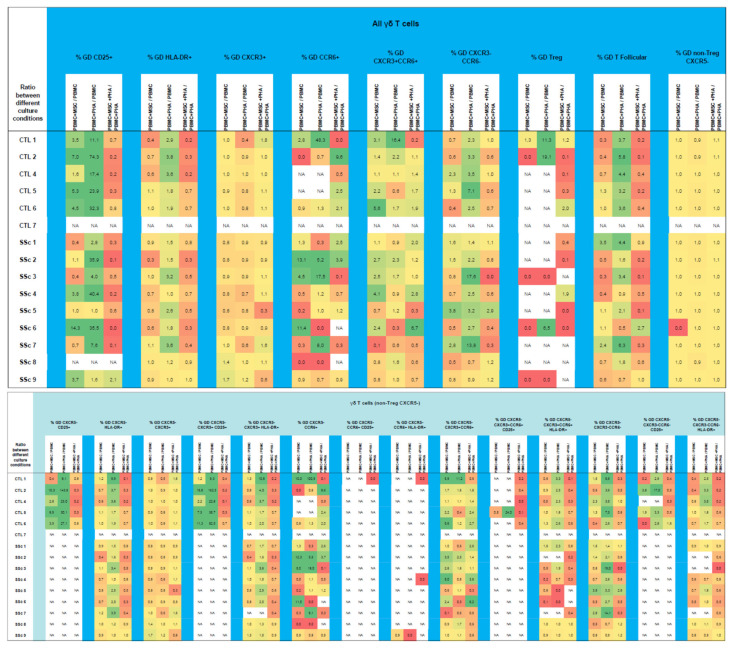
Effect of MSCs’ and PHA stimulation on the polarization and activation of γδ T cells isolated from HC and SSc patients. Results are presented as a ratio between PBMC + MSC/PBMC (indicating the effect of MSCs on unstimulated PBMCs), PBMC+PHA/PBMC (to assess the effect of PHA on PBMCs), and PBMC + MSC + PHA/PBMC + PHA (to evaluate the effect of MSCs on PHA-stimulated PBMCs) for each participant. Yellow denotes no effect or minimal effect (ratio ≈ 1), orange to red indicates inhibition (ratio ≤ 0.7), and green represents induction/stimulation (ratio ≥ 1.5). HC, healthy controls; MSCs, mesenchymal stromal cells; PBMCs, peripheral blood mononuclear cells; PHA, phytohemagglutinin; SSc, systemic sclerosis.

**Table 1 biomedicines-11-01329-t001:** Demographic and clinical data of SSc patients.

Patient	Gender	Age	SSc Subtype	Disease Duration (Years)	Auto-Antibody Profile	Organ Involvement	Digital Ulcers	mRSS	SSc Treatment	SSc Conditions for Treatment	Comorbidities and Treatment
SSc 1	F	57	dcSSc	5	anti-Scl-70	None	No	4	Mycophenolate mofetil, Prednisolone	Skin involvement	Breast carcinoma: Anastrozole Dyslipidemia: Atorvastatin, Ezetimibe
SSc 2	F	33	dcSSc	4	anti-Scl-70	Lung	Yes	18	Mycophenolate mofetil, Nifedipine	Lung involvement, Raynaud phenomenon, digital ulcers	Depression: Fluoxetine
SSc 3	F	72	lcSSc	7	ACA	Heart	No	10	Bosentan, Sildenafil	Pulmonary arterial hipertension	Hypothyroidism: Levothyroxine Depression: Mirtazapine, Quetiapione, Alprazolam
SSc 4	F	36	lcSSc	5	anti-Scl-70	None	No	0	Pentoxifylline, Nifedipine	Raynaud phenomenon	None
SSc 5	F	69	lcSSc	10	ACA	None	No	0	Pentoxifylline	Raynaud phenomenon	Depression: Escitalopram Osteoporosis: Alendronic acid+Calcium +VitaminD Dyslipidemia: Rosuvastatin Arrhythmia: Bisoprolol
SSc 6	F	40	lcSSc	5	anti-Scl-70	None	No	10	Methotrexate, Prednisolone, Amlodipine	Skin involvement, Raynaud phenomenon	Depression: Trazodone, Paroxetine
SSc 7	F	64	lcSSc	5	ACA	None	Yes	6	Pentoxifylline, ASA 100	Raynaud phenomenon, digital ulcers	Cardiac insufficiency: Furosemide, Sacubitril+Valsartan, Bisoprolol, Spironolactone, Ivabradine, ASA 100 Gastric Ulcer: Pantoprazole Depression: Escitalopram, Zolpidem
SSc 8	F	82	lcSSc	9	ACA	None	Yes	6	Azathioprine, Pentoxifylline	Raynaud phenomenon	Dyslipidemia: Rosuvastatin Cardiac insufficiency: Furosemide, Valsartan, Spironolactone Osteoporosis: Alendronic acid+Calcium +VitaminD
SSc 9	F	62	dcSSc	3	anti-Scl-70	Lung	Yes	21	Mycophenolate mofetil, Bosentan, ASA 100, Nifedipine	Lung involvement, Raynaud phenomenon, digital ulcers	Osteoporosis: Alendronic acid+Calcium +VitaminD Depression: Venlafaxine, Alprazolam

ACA, anti-centromere antibodies; anti-Scl-70, anti-topoisomerase I antibody; ASA 100, acetylsalicylic acid 100 mg; F, female; lcSSc, limited cutaneous SSc; dcSSc, diffuse cutaneous SSc; mRSS, modified Rodnan skin score [49]; SSc, systemic sclerosis. In the Comorbidities and Treatment column, the comorbidity is underlined and followed by the respective treatment.

**Table 2 biomedicines-11-01329-t002:** Monoclonal antibody panel for assessing T cell subsets and their activation statuses, indicating the clone and commercial source.

Fluorochrome	FITC	PE	PerCP-Cy5.5	PC7	APC	Alexa Fluor 700	APC Alexa 750	V450	BV510	BV605
Antibody marker	TCRγδ	CXCR3	CCR6	CD25	CD4	CD8	CD3	HLA-DR	CD127	CXCR5
Clone	IMMU 510	1C6	11A9	B1.49.9	13B8.2	B9.11	UCHT1	L243	HIL-7R-M21	J252D4
Commercial source	Beckman Coulter	BD Pharmingen	BD Pharmingen	Beckman Coulter	Beckman Coulter	Beckman Coulter	Beckman Coulter	BD	BD	BioLegend

Abbreviations: APC, allophycocyanin; BV, brilliant violet; FITC, fluorescein isothiocyanate; PE, phycoerythrin; PC7, phycoerythrin-cyanine 7; PerCP-Cy5.5, peridinin chlorophyll protein cyanine 5.5. Commercial sources: BD (Becton Dickinson Biosciences, San Jose, CA, USA); BD Pharmingen (San Diego, CA, USA); Beckman Coulter (Miami, FL, USA); BioLegend (San Diego, CA, USA).

**Table 3 biomedicines-11-01329-t003:** Immunomodulation by MSCs. Proportion of T cell subsets whose activation or polarization was modulated by mesenchymal stromal cells (MSCs).

	Number of T Cell Subsets Whose Early Activation (CD25^+^) is Downregulated by MSCs	Number of T Cell Subsets Whose Late Activation (HLA-DR^+^) Is Downregulated by MSCs	Number of T Cell Subsets Whose (Early or Late) Activation Is Downregulated by MSCs	Number of T Cell Subsets Whose Polarization is Modulated by MSCs
HC				
CD4^+^ T cells	5 out of 8	11 out of 16	11 out of 16	10 out of 18
CD8^+^ T cells	3 out of 7	6 out of 9	6 out of 9	3 out of 14
CD4^+^CD8^+^ T cells	1 out of 5	3 out of 7	3 out of 7	3 out of 12
CD4^−^CD8^−^ T cells	2 out of 6	4 out of 6	5 out of 6	3 out of 10
γδ T cells	2 out of 4	4 out of 4	4 out of 4	3 out of 10
Total (HC)	13 out of 30	28 out of 42	29 out of 42	22 out of 64
SSc				
CD4^+^ T cells	4 out of 8	10 out of 16	10 out of 16	10 out of 18
CD8^+^ T cells	2 out of 7	6 out of 9	7 out of 9	1 out of 14
CD4^+^CD8^+^ T cells	2 out of 5	0 out of 7	2 out of 7	2 out of 10
CD4^−^CD8^−^ T cells	1 out of 5	3 out of 5	3 out of 5	0 out of 10
γδ T cells	1 out of 1	4 out of 4	4 out of 4	0 out of 6
Total (SSc)	10 out of 26	23 out of 41	26 out of 41	13 out of 58

HC, healthy controls; MSCs, mesenchymal stromal cells; SSc, systemic sclerosis.

**Table 4 biomedicines-11-01329-t004:** Differences between SSc patients presenting ACA (*n* = 4) and anti-Scl-70 (*n* = 5) autoantibody profiles concerning the activation and polarization of distinct T cell subsets measured at basal levels (i.e., PBMCs cultured alone for 4 days).

	HC (*n* = 6) Mean ± Standard Deviation	ACA SSc (*n* = 4) Mean ± Standard Deviation	Anti-Scl-70 SSc (*n* = 5) Mean ± Standard Deviation	*p* Value (ACA vs. Anti-Scl-70)
CD4^+^ T cells				
% Total CD4^+^ CD25^+^ T cells	4.53 ± 1.53	19 ± 5.74	10 ± 4.75	*p* < 0.07
% Th17 CD25^+^	21 ± 6.51	29 ± 4.93	18 ± 4.00	*p* < 0.05
% Th CXCR3^−^CCR6^−^ CD25^+^	3.85 ± 1.06	19 ± 7.83	10 ± 3.20	*p* < 0.05
% Tfh CD25^+^	17 ± 6.47	35 ± 14.6	21 ± 4.66	*p* < 0.07
% Tfh CXCR3^−^CCR6^−^ CD25^+^	20 ± 6.41	31 ± 6.73	23 ± 3.58	*p* < 0.07
CD8^+^ T cells				
% Tc1	69 ± 8.20	73 ± 6.81	51 ± 14.5	*p* < 0.05
% Tc CXCR3^−^CCR6^−^	29 ± 7.80	24 ± 4.08	45 ± 16.8	*p* < 0.05
% CD8^+^ Tf CXCR3^−^CCR6^−^	9.45 ± 2.47	6.14 ± 4.39	15 ± 4.85	*p* < 0.05
CD4^+^CD8^+^ T cells				
% CD4^+^CD8^+^ Treg cells	1.09 ± 0.65	0.27 ± 0.16	2.10 ± 1.89	*p* < 0.05
CD4^−^CD8^−^ T cells				
% CD4^−^CD8^−^ CXCR3^−^CCR6^−^ HLA-DR^+^	32 ± 18.4	78 ± 16.1	43 ± 15.3	*p* < 0.07
% CD4^−^CD8^−^ Tf CXCR3^−^CCR6^−^	22 ± 2.86	11 ± 10.5	30 ± 15.0	*p* < 0.07

ACA, anti-centromere antibody; anti-Scl-70, anti-topoisomerase I antibody; HC, healthy controls; SSc, systemic sclerosis. Mann–Whitney test was performed to compare ACA vs. anti-Scl-70 SSc patients. Differences were considered statistically significant at *p* < 0.05.

## Data Availability

The data presented in this study are available in this article and its Appendix A.

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
