# Peer review of "Umbilical-Cord-Derived Mesenchymal Stromal Cells Modulate 26 Out of 41 T Cell Subsets from Systemic Sclerosis Patients"

_biomedicines, 2023, doi:10.3390/biomedicines11051329_

Round 1
Reviewer 1 Report
The study entitled Umbilical cord-derived mesenchymal stromal cells modulate 26 out of 41 T cell subsets from systemic sclerosis patients, brings new inside in the stem cells treatment for various disease (systemic sclerosis in this manuscript). Far from being elucidated, the mechanisms by which stem cells can be beneficial require detailed investigations.
The following observation have to be made:
Introduction
- line - 52-56 - please insert the references for your affirmations in this paragraph. There are to many details written and only one reference in the en (ref -3).
Please describe more details about the endothelial cells death in systemic sclerosis. Since the vasculitis is one of the primary pathological process in SS, the endothelial cells death mechanism is important.
- line 63-64 - please insert the role of matrixmetalloproteinases (MMP) in extracelular matrix involvement in SS pathogenesis.
- line 119 - please explain more clear the aim of the study because is not clear where from are MSC you used in this study.. Please explain here the abbreviation of PBMC, because here you used the first time this abbreviation.
Mt and method
Please describe the period of time you enrroled the patients (month/year).
Please indicate the comorbidities of the patients, because in table 1 are various medication as Nifedipine, Pentoxiphyline, Amplodipine which have been recommended for various diseases.
Fig 2 - please split the information in this figure in 2 ore more figures because the text is to small to be read properly.
Conclusions - please mention there are more studies needed until the method to be translated to clinical studies because the number of the patients in this study was small.
Please mention the limitation of the study - eventually the small group of patients.
Reviewer 2 Report
In this article, Laranjeira et al examined peripheral blood mononuclear cells from healthy individuals and SSc patients, and they were co-cultured with umbilical cord-derived mesenchymal stromal cells (MSCs) to assess how MSCs affected the activation and polarization of T cell subsets, including Th1, Th17 and Treg. MSCs downregulated the activation or polarization of most T cell subsets. The increased activation status in SSc patients were all downregulated by MSCs. They provided a wide-ranging vision about how MSCs affect T cells. This is a very valuable study that suggests the usefulness of MSCs in the treatment of SSc. However, I have some questions and other concerns, so I would like to ask the following questions.
major concerns)
1) In Figures 4 and 6, this study discusses the % throughout. However, the % could change if other cells increase, and the % could remain the same if the entire population multiplies. Therefore, the absolute number (n) should be used to indicate the number of cells. In addition to the % notation, please add a notation using absolute numbers.
2) The results of this study are very valuable in suggesting the pathogenic role of activated T cells and IL-17-producing T cells in systemic sclerosis. And on the other hand, there are clinical trials and other studies that suggest the importance of B cells in clinical research (Safety and efficacy of rituximab in systemic sclerosis (DESIRES): a double-blind, investigator-initiated, randomised, placebo-controlled trial. Ebata, Satoshi et al. The Lancet Rheumatology, Volume 3, Issue 7, e489 - e497). Basic studies also suggest that B cells induce T cell differentiation through cytokine production, and are involved in pathogenicity (Fukasawa T, et al. Single-cell-level protein analysis revealing the roles of autoantigen-reactive B lymphocytes in autoimmune disease and the murine model. Elife. 2021 Dec 2;10:e67209. doi: 10.7554/eLife.67209. PMID: 34854378; PMCID: PMC8639144.). In addition, recent results of clinical trials for systemic sclerosis suggest that brodalumab, an anti-IL-17RA receptor antibody, could be important in the pathogenesis of various forms of systemic sclerosis, such as skin sclerosis and pulmonary fibrosis (Fukasawa T, et al. Interleukin-17 pathway inhibition with brodalumab in early systemic sclerosis: analysis of a single-arm, open-label, phase 1 trial. Journal of the American Academy of Dermatology. 2023.). Based on and citing these references, please also provide a Discussion on the importance of IL-17 in T cells.
3) Related to the above, MSCs have been shown to suppress activated Th17 cells and other cells (Figure 4). What is the possible mechanism of such suppression by MSCs? Is this suppression of T cells mediated by some substance produced by MSCs? Or is it a direct interaction?
4) In addition, although this is an in vitro experiment, can MSCs be used to suppress skin sclerosis and pulmonary fibrosis in mouse models of systemic sclerosis?
minor concerns)
1) In references section, the ref number is double numbered. Please correct this as only one notation is sufficient.
Author Response
"Please see the attachment."

Reviewer 3 Report
This manuscript describes the impact of MSCs on the phenotype and function of several T cell subsets from HS versus 9 SSc patients (4 patients with ACA and 5 patients with anti-Scl70 Abs). The source of MSCs used in this study was SLCTmsc02, an umbilical cord tissue cell suspension cryopreserved. The authors observed that MSCs affected the activation and polarization of several T cell subsets.
Here are some concerns:
- Figure 2 lacks visibility and needs to be modified and in general, all the figures are really difficult to read and should be modified.
- SScs patients with ACA and anti-Scl70 Abs have very different clinical phenotypes. Did the authors observe any differences within these two categories of patients? This deserves to be discussed in the manuscript.
Reviewer 4 Report
Systemic sclerosis (SSc) is an immune-mediated disease, wherein T cells are particularly implicated, with poor prognosis and limited therapeutic options. Given the immunomodulatory, anti-fibrotic and pro-angiogenic potential of MSCs, the authors reasoned that mesenchymal stem/stromal cell (MSC)-based therapies can be beneficial to SSc patients.They collected, peripheral blood mononuclear cells from healthy individuals (HC, n=6) and SSc patients (n=9) and co-cultured with MSCs, in order to assess how MSCs affected the activation and polarization of 58 different T cell subsets, including Th1, Th17 and Treg. They report that the increased activation status some SSc T cell subsets displayed, were all downregulated by MSCs. This si a nice in vitro work which could serve as a basis for future in vivo work. This shall be acnowledged as a study limitation. The authors shall also note that MSCs have been recently used to limit neuroinflammation in a vascular disease of the brain (see, doi: 10.1007/s00395-021-00881-9)
Round 2
Reviewer 2 Report
The authors have adequately answered the questions. No additional comments.
Reviewer 3 Report
The authors have satisfactorily addressed my concerns.
Reviewer 4 Report
The authors have successfully addressed my concerns.